# Choice of Assessment and Subsequent Risk of Breast Cancer among Women with False-Positive Mammography Screening

**DOI:** 10.3390/cancers15061867

**Published:** 2023-03-20

**Authors:** Bayan Sardini, Susanne Fogh Jørgensen, Lisbet Brønsro Larsen, Mohammad Talal Elhakim, Sisse Helle Njor

**Affiliations:** 1Department of Public Health Programmes, University Research Clinic for Cancer Screening, Randers Regional Hospital, Skovlyvej 15, 8930 Randers, Denmark; 2Department of Clinical Medicine, Aarhus University, Palle Juul-Jensens Blvd. 82, 8200 Aarhus, Denmark; 3Department of Radiology, Odense University Hospital, University of Southern Denmark, 5000 Odense, Denmark

**Keywords:** breast cancer, screening, false-positive, mammography, ultrasound

## Abstract

**Simple Summary:**

Women with one or more false-positive results have a two- to four-fold higher risk of subsequently developing breast cancer than women with negative screening results do. This study investigated whether subsequent breast cancer risks are different for different choices of assessments performed after a positive breast cancer screening. This register-based national study showed that women who had an assessment including ‘only ultrasound’ or ‘only additional mammography’ had higher relative risks (RR) of next-round screen-detected cancer of 1.52 (95% CI: 0.93–2.47) and 1.67 (95% CI: 0.54–5.16) compared to those of women who underwent assessments with ultrasound and additional mammography. Invasive assessments (i.e., assessments that included biopsy) that lacked an ultrasound or additional mammography were not associated with an increased risk of subsequently developing cancer, leaving the assessments ‘all three elements’, ‘only additional mammography’, or ‘only ultrasound’ with the highest subsequent risks of breast cancer. It might therefore be important to remind women who are assessed with the above-mentioned assessments to attend subsequent screenings.

**Abstract:**

Women with false-positive mammography screening results have a two- to four-fold higher risk of breast cancer. This study aimed to investigate if the subsequent risk of breast cancer after a false-positive mammography screening is associated with the received diagnostic assessment. The study population consisted of women who underwent false-positive mammography screening from January 2010 to June 2019. They were categorised into seven groups depending on the elements in the assessment (standard care: additional mammography, ultrasound, and if they had a relevant biopsy). Risks of interval cancer, next-round screen-detected cancer, and long-term breast cancer for non-standard care assessments were compared to standard care assessments using Binomial and Cox regression models. We included 44,279 women with a false-positive result. Invasive assessments that lacked an ultrasound or additional mammography were not more associated with an increased risk of subsequent cancers compared to that of ‘all three elements’. The few assessments that included ‘only ultrasound’ or ‘only mammography’ resulted in higher relative risks of next-round screen-detected cancer of 1.52 (95% CI: 0.93–2.47) and 1.67 (95% CI: 0.54–5.16), respectively, compared to that of standard care. The increased subsequent risk of breast cancer among women with a previous false-positive result was not found to be correlated with the choice of elements in the assessment process.

## 1. Introduction

Several studies have shown that women with a previous false-positive result at a mammography screening have an increased risk of being subsequently diagnosed with breast cancer [1,2,3,4,5,6]. Women who have had one false-positive mammography screening result have a two-fold higher risk of being subsequently diagnosed with breast cancer, while women who have had two false-positive mammography screening results have a four-fold higher risk compared to those of women who have only had negative mammography screening results [4]. A Spanish study even estimated that there is a nine-fold increased risk of being subsequently diagnosed with breast cancer among women whose mammography features changed between two subsequent false-positive mammography screening results compared to those of women who have only had negative mammography screening results [2].

Women who have previously received a false-positive result at a mammography screening could be diagnosed either with breast cancers that have emerged after the false-positive screening or breast cancers that were missed during the screening, i.e., misclassification [1,6]. Von Euler-Chelpin et al. showed that the increased risk of being diagnosed with a subsequent breast cancer after a false-positive result cannot solely be explained by misclassification [3]. The increased risk of being subsequently diagnosed with breast cancer among women with a false-positive result is, therefore, probably due to both misclassifications and underlying biological susceptibility.

Misclassification, i.e., missed cancer diagnoses, could happen because of insufficient diagnostic assessments after the positive screening examination. European Guidelines for breast cancer screening and diagnosis [7] recommends ‘triple assessment’ after positive mammography screening, including breast examinations, additional mammographic imaging, and ultrasound examination. Furthermore, if it is relevant, biopsy of the detected lesion should be performed [7,8,9]. Even though previous research shows that diagnostic assessment often happens in accordance with the recommendations, not all of these procedures are performed during the diagnostic assessment [10].

To reduce the subsequent risk of breast cancer among women with a false-positive result, it is important to know if the increased risk of breast cancer is associated with the assessment that led to the false-positive mammography screening result. If the subsequent risk of being diagnosed with breast cancer is more frequent after a specific diagnostic assessment, relevant focus can be placed on decreasing the high breast cancer risk after a false-positive result.

This study aimed to investigate whether women who had a false-positive mammography screening and did not receive the recommended assessment as per the guidelines had an increased risk of being subsequently diagnosed with breast cancer.

## 2. Materials and Methods

### 2.1. Setting

Nation-wide mammography screening offering biennial mammography screening to women aged 50–69 years started in Denmark in 2007 [11]. Five Danish regions administer the screening programme according to national guidelines for mammography screening [12].

All mammograms are read independently by two radiologists, who determine the normality/abnormality of the screening mammograms by consensus. Cases of disagreement between the readers are resolved either by a group consensus or through arbitration by a single senior radiologist. Women with abnormal results are offered a diagnostic assessment according to the national clinical guidelines for assessment, i.e., an ultrasound, an additional mammography, and if it is relevant, a biopsy (standard of care) [7,12]. This assessment has to start no more than 14 days after the woman receives the abnormal screening result [12]. After the assessment, women with false-positive results return to the screening programme [12].

### 2.2. Study Design and Population

Although the Danish screening age targets women aged 50–69 years, some women are invited just before turning 50 years old, and some have their last invitation at the age of 71 [13,14]. The study population, therefore, consisted of all women aged 49–71 years who had a false-positive result following a mammography screening invitation between 1 January 2010 and 30 June 2019. Women who had previously been diagnosed with breast cancer were excluded from the study population as they have a higher risk of interval cancers than women without a previous breast cancer diagnosis do [15]. As we also expect that the majority of women with previous breast cancer would have received the most comprehensive follow-up, i.e., standard of care assessment, this would have biased our results, since including them could increase the risk of subsequent cancer in the standard of care assessment group. Furthermore, women who emigrated or died within 30 days after the screening date were excluded. The study population was followed for breast cancer diagnoses until death, emigration, or the end of follow-up (31 December 2021).

### 2.3. Data

The study population was identified based on information on invitations from the Danish Quality Database for Mammography Screening (DKMS) [11]. DKMS received this information directly from the regions in charge of inviting the target population.

Screening results and data on the performed assessment were retrieved from the Danish National Patient Register and the National Pathology Register. Data on cancer diagnoses were obtained from the Danish Cancer Register and the National Pathology Register. Data on age at screening as well as dates of death and emigration were retrieved from the Civil Registration System. All the data were merged using the unique Danish personal identification number issued to all Danish residents at birth or immigration [16,17]. All registers were deemed to have high completeness and reliability [11,16,17,18,19,20,21].

### 2.4. Definitions

The study population was initially categorised into invasive or non-invasive diagnostic assessment depending on whether a biopsy was performed in the process. Furthermore, the diagnostic assessment was categorised into groups based on the specific procedures performed after the positive screening result, i.e., additional mammography (including tomosynthesis and breast MRI), ultrasound examination, biopsy (including core and fine-needle biopsies), and combinations of them. The invasive diagnostic assessments were thereby grouped into: standard of care, i.e., additional mammography, ultrasound, and biopsy (group 1); other assessment, i.e., assessment including only biopsy (group 2); assessment including biopsy and additional mammography (group 3); assessment including biopsy and ultrasound (group 4). Similarly, the non-invasive diagnostic assessments were grouped into standard of care, i.e., additional mammography and ultrasound (group 5), assessment only including ultrasound (group 6), and assessment only including additional mammography (group 7). Assessment procedures performed more than 6 months after the screening assessment were not considered to be part of the diagnostic assessment following a positive mammography screening.

Since we have no information on whether cancers following an abnormal screening are screen-detected or interval cancers, we defined a breast cancer diagnosed after an abnormal screening as a screen-detected one if the breast cancer was diagnosed 0–180 days after the screening date. Breast cancers diagnosed between 180 days and two years/invitation to the next screening round (whatever came first) were defined as interval cancers.

The ICD10 code C50 from the Cancer Registry was used to exclude women with previously diagnosed breast cancer before their invitation and to identify breast cancer diagnoses after a false-positive screening [19]. The Cancer Registry was only updated to include cancers diagnosed before 2020, therefore, cancers diagnosed in 2020 and 2021 were identified using the Danish Standard Nomenclature of Medicine (SNOMED) in the Pathology Register. All histological samples taken from the breast with the morphological code for cancer were included [21].

### 2.5. Analyses

To investigate the risk of misclassification, we calculated for each assessment group (a) the risk of developing an interval cancer in the two years following the false-positive screening and (b) the risk of developing a screen-detected cancer that is diagnosed at the successive screening (next round). Furthermore, to investigate how different choices of assessment correlate with the subsequent risk of breast cancer, we calculated for each assessment group (c) the long-term risk of developing breast cancer any time before end of follow-up, i.e., including both a, b, and breast cancers diagnosed due to symptoms or at later screening examinations.

The risk ratios of interval cancer or next-round screen-detected cancer for the different assessment groups compared to those of the relevant standard of care (i.e., additional mammography, ultrasound, and biopsy, or additional mammography and ultrasound) were estimated using a binominal regression and adjusted for age. The hazard ratio (HR) of developing long-term breast cancer at any time before the end of follow-up was estimated using Cox regression. A test for the interaction between age groups and assessment groups was performed in all three models. The interaction terms were then dropped, and the models included the age groups only together with the assessment groups. The analyses were performed using the statistical software STATA, version 17.0 [22].

## 3. Results

### 3.1. Study Population Characteristics

Following invitation to the Danish mammography screening programme between 1 January 2010 and 30 June 2019, 2,549,722 mammography screenings were performed. Among these, 61,081 (2.4%) women had an abnormal mammography screening (Figure 1). After the exclusion of women younger than 49 and those older than 71, women who died within 30 days after screening, women with a previous breast cancer diagnosis, and women with a true-positive screening, our study population consisted of 44,279 women with a false-positive mammography screening result. The risk of developing an interval cancer or a long-term breast cancer was estimated in the entire study population, while the risk of developing a next-round screen-detected cancer was only estimated among those 38,684 women who participated in a successive mammography screening and had no cancer diagnosis before this invitation (Figure 1).

The number and incidence of interval cancers, next-round screen-detected cancers, and long-term breast cancers until the end of 2021 are presented in Table 1 for the various assessment categories. The most frequent assessment, i.e., additional mammography and ultrasound, was received by 68.3% (30,223) of the study group, and among those, 26,655 (68.9%) participated in a successive screening. Only 2.7% received no assessment, but 917 of those did participate in a successive screening (76%). The survival analysis for long-term breast cancer shows that 2020 (4.6%) were censored due to death and none were censored due to emigration.

### 3.2. Risk of Cancer

The adjusted relative risk (RR) of having an interval cancer was similar or lower in all of ‘the invasive diagnostic assessments’ groups, i.e., group 2 ‘only biopsy’, group 3 ‘biopsy and additional mammography’, and group 4 ‘biopsy and ultrasound’, compared to that of the standard of care group (women having ‘all three elements’): RRs of 0.62 (95% CI: 0.09–4.41), 1.09 (95% CI: 0.15–7.72), and 0.32 (95% CI: 0.05–2.32), respectively (Table 2).

The adjusted relative risks of having a next-round screen-detected cancer were almost similar in the ‘invasive diagnostic assessments’ groups, i.e., groups 2 and 4, with an RR of 0.98 (95% CI: 0.24–3.92) and an RR of 1.22 (95% CI: 0.50–2.96), respectively (Table 3). The invasive diagnostic assessment groups 2–4 had a lower HR of having a long-term breast cancer compared to that of the ‘all three elements’ group HR: 0.72 (95% CI: 0.32–1.62), 0.76 (95% CI: 0.28–2.02), and 0.75 (95% CI: 0.44–1.27) (Table 4), respectively.

However, the results did not show the same pattern among the non-invasive groups. Compared to group 5 ‘ultrasound and additional mammography’, both the ‘only ultrasound’ and the ‘additional mammography screening’ groups had a higher risk of next-round screen-detected cancer, 1.52 (95% CI: 0. 93–2.47) and 1.67 (95% CI: 0. 54–5.16), respectively (Table 3).

By comparing the group that only lacked a biopsy, i.e., ‘ultrasound and additional mammography’ to the ‘all three elements’ group, the results showed significantly lower risks of developing an interval cancer, an RR of 0.66 (95% CI: 0.51–0.86) and a next-round screen-detected cancer RR of 0.78 (95% CI: 0.62–0.98). The risk of developing long-term breast cancer with a follow-up up of 12 years had a non-significantly lower HR of 0.92 (95% CI: 83–1.02) (Table 2, Table 3 and Table 4).

The relative risks of having a next-round screen-detected cancer and a long-term breast cancer were higher, an RR of 1.50 (95% CI: 0.89–2.52) and an HR of 1.29 (95% CI: 1.01–1.64), respectively, in group 0 ‘no assessment’ compared to that of the reference group ‘all three elements’ (only crude numbers are shown in Table 1).

In all three analyses, the relative risk differences found between the assessment groups were not significantly different across age groups (no significant interaction between age and assessment groups). Furthermore, the risk directions in all three outcomes, i.e., interval cancer, next-round screen-detected cancer, and long-term breast cancer, were the same when they were adjusted for age.

## 4. Discussion

### 4.1. Main Findings

This study showed that women who received an invasive diagnostic follow-up without all three elements (i.e., additional mammography, ultrasound, and biopsy) had a lower or similar risk of interval cancer, next-round screen-detected cancer, and long-term breast cancer compared to those of the women who received all three elements. Therefore, missing one or two elements other than the biopsy does not seem to result in an increase in the rate of missed cancers. Women who had ‘only an ultrasound’ or ‘only an additional mammography’ during a non-invasive assessment had a non-statistically significant higher risk of next-round screen-detected cancers compared to those of the women who received both assessments combined; hence, missing one of the elements might be related to a missed cancer. Although the statistical significance is absent in the results due to small numbers in some groups, clinical significance may be present. We therefore reported results as point estimates with 95% confidence interval estimates as recommended by STROBE and others [24,25,26].

### 4.2. Strength and Weakness

The registers used for data retrieval are high-quality, reliable Danish registers covering all residents, with very few missing data. In addition, DKMS obtained data directly from the regions responsible for inviting the target population, hence the quality is considered to be high.

This enabled us to track detailed assessment pathways on an individual level with small losses to follow-up for all women with an abnormal screening mammogram. However, the study has some limitations. Some women in the no assessment and only biopsy groups might have undergone a diagnostic assessment or diagnostic imaging procedures in private hospitals or abroad. As most data from private hospitals are available in the registers and the clinical recommendations state that assessment procedures should be performed in Danish public Breast Centres [16], unless the expected waiting time exceeds 28 days, we expect this problem to be small [12,16,17]. Immigrants might prefer to have an assessment in their home country and would, therefore, be classified as having had no assessment in our study. There were, however, only five immigrants in the no assessment group, therefore, this can only have had a minor effect on our results for the ‘no assessment’ group.

In the analysis of risk of interval cancers, we did not exclude women who died in this period, therefore, the interval cancer risk has been slightly underestimated. As very few women died in this period, this can only have had a minor effect on our result and cannot have changed the overall conclusions.

As we did not know whether breast cancer diagnosed within two years after a screening was an interval cancer or late screen-detected cancer, we had to assume that those diagnosed within six months of the screening were screen-detected cancers and the remainder were interval cancers. Therefore, we might have misclassified a few screen-detected cancers as interval cancers, and vice versa. Previous studies have shown that the misclassification rate is only 0.1%, therefore, this can only have had a minor effect on our results [15].

The choice of assessment, as well as the subsequent risk of breast cancer, might be different for incidentally and prevalently screened women. However, as our data do not include the prevalent round of the Danish mammography screening programme, most prevalent screenings are performed on women aged 50–51 years, with a limited number of breast cancer cases. For instance, our study population contained only 655 prevalently screened women who also had another screening, at which five breast cancers were detected. These low numbers among prevalently screened women did not affect our results notably.

Finally, according to the guidelines, examinations such as tomosynthesis and MRI fall under the category ‘supplementary mammographic examinations’. As these procedures have a higher sensitivity in detecting breast cancer, it would have been interesting to look at them separately. However, this was not possible due to the small number of cases.

### 4.3. Other Studies

A Spanish study showed that the relative risk for subsequent cancers (>2 year after screening) in the non-invasive group, which was compared to that of the invasive group, was 0.70 (3.1 vs. 4.4). This is in line with our similar adjusted relative risk for screen-detected breast cancer: 0.78 (0.62–0.98). The unadjusted relative interval cancer risk among women with a false-positive screening was 0.16 (1.9 vs. 12.0) for non-invasive assessments, which was compared to that of women who underwent invasive assessments [27]. This is lower than our finding of the adjusted relative risk of interval cancers in the ‘ultrasound and additional mammography’ (i.e., non-invasive) group: 0.66 (0.51–0.86), which was compared to that of the ‘all three elements’ (i.e., invasive) group. The lower relative interval cancer risk is probably due to a higher diagnostic referral rate in Spain (5.1%) than that in Denmark (2.4%) [6,10]. In addition, our results confer with the Spanish results, where women referred to assessment including a biopsy had a higher risk of breast cancer, as these women referred for a biopsy, were more likely to have a lesion. A study from the US found the subsequent risk of breast cancer after a false-positive screening to be 5.51% for those who were recommended additional mammograms and 7.01 for those who were recommended an additional biopsy. In our study, the long-term risk of breast cancer was also somewhat lower for those who were recommended biopsy (4.1%) than that among those who were recommended additional mammography (4.9%) (from Table 1) [28].

Finally, our findings that missed elements in the assessment cannot solely explain the higher risk of breast cancer in women with false-positive mammograms are in line with a previous Danish study [15].

### 4.4. Clinical Implications

The higher risk of interval cancer and long-term breast cancer among women whose assessment included all three elements shows that a missing element in the choice of assessment cannot explain the subsequent risk of breast cancer. Nevertheless, the diagnostic assessment including ‘only an ultrasound’ or ‘only an additional mammography’ might not serve as an appropriate substitute for the diagnostic assessment including both an ‘additional mammography and ultrasound’, as this could entail breast cancers being missed. Moreover, other underlying reasons for the choice of examination in the assessments may be influenced by other factors, such as suspicious clinical findings or particular mammographic features, which should be considered when one is assessing the extent of confounding in further studies, but this was beyond the scope of our study.

## 5. Conclusions

The risk of interval cancer, next-round screen-detected cancer, and long-term cancer in women who underwent false-positive mammography screening was not higher in groups with varying combinations of assessments compared to that of the standard of care group. Thus, the increased subsequent risk of breast cancer among women with a previous false-positive result may not be explained by the choice of elements in the assessment process. Nevertheless, ‘only an ultrasound’ and ‘only an additional mammography’ assessments do not seem to be an acceptable alternative to a ‘mammographic plus ultrasound assessment’, and they might even be associated with missed breast cancers. It underscores the importance of participation for false-positive women in subsequent invitations.

## Figures and Tables

**Figure 1 cancers-15-01867-f001:**
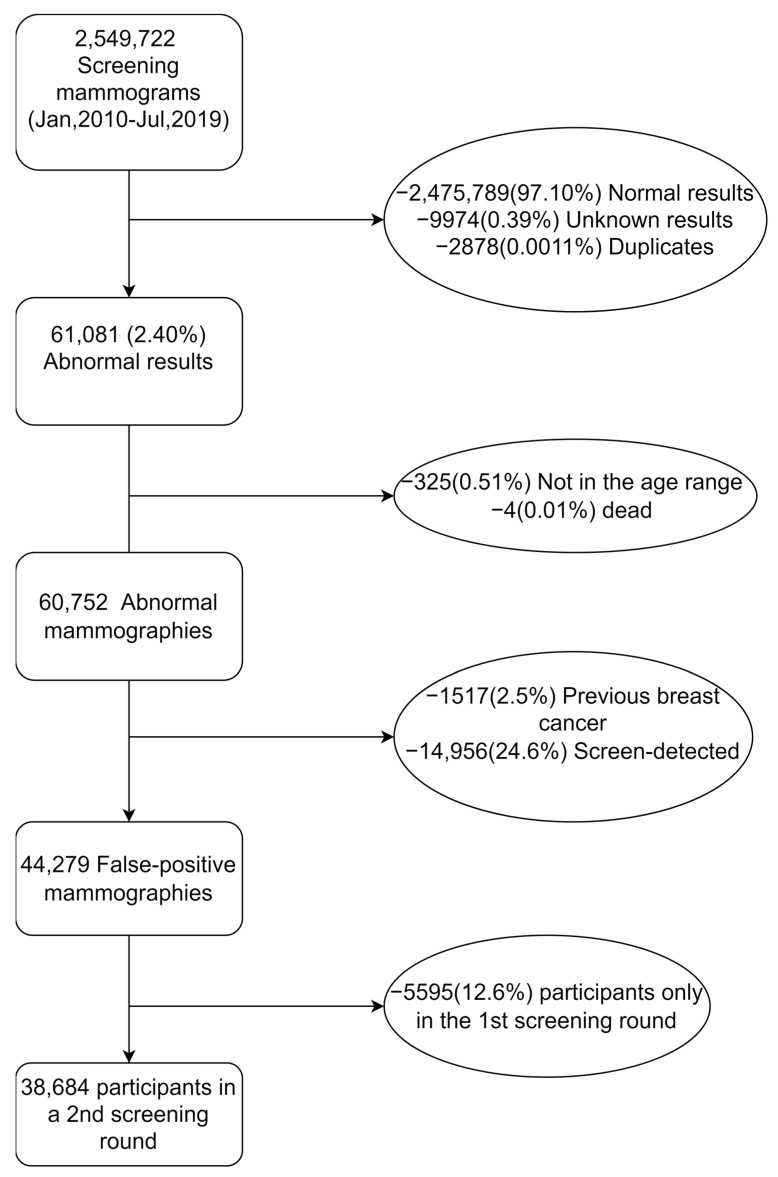
Flowchart of inclusions and exclusions in the study population.

**Table 1 cancers-15-01867-t001:** The characteristics of the study population by diagnostic assessment procedures performed.

		Age	Interval Cancers			Next-Round Screen-Detected Cancer		Long-Term Breast Cancers until End of 2021		
		Mean (SD)	*N*** (%)	CI (% of *N*)	Mean (SD) Time to IC in Years	*N** (%)	CI (% of *N*)	*N*** (%)	CI (% of *N*)	Mean (SD) Follow-up Time to BC in Years
	Total	57.8 (6.3)	44,279	<264 (<0.6)	1.4 (0.4)	38,684	<401 (<1.0)	44,279	<1827 (<4.2)	11.6 (1.4)
	Group 0: No assessment	58.5 (6.4)	1211 (2.7)	8 (0.7)	1.2 (0.4)	917 (2.4)	16 (1.7)	1211 (2.7)	74 (6.1)	11.3 (2.0)
Invasive diagnostic assessments	Group 1: All three elements (standard of care)	57.4 (6.4)	10,612 (24.0)	84 (0.8)	1.1 (0.4)	9195 (24.0)	107 (1.2)	10,612 (24.0)	526 (5.0)	11.6 (1.5)
	Group 2: Only biopsy	57.7 (6.6)	685 (1.6)	<5 (<0.7)	NA	599 (1.6)	7 (1.2)	202	6 (3.5)	11.6 (1.4)
	Group 3: Biopsy and additional mammography	58.0 (6.3)						113	<5	11.7 (1.4)
	Group 4: Biopsy and ultrasound	58.4 (6.3)						370	14	11.5 (1.7)
Non-invasive diagnostic assessments	Group 5: Ultrasound and additional mammography (standard of care)	57.8 (6.2)	30,223 (68.3)	160 (0.5)	1.2 (0.4)	26,655 (68.9)	249 (0.9)	30,223 (68.3)	1141 (3.8)	11.7 (1.3)
	Group 6: Only ultrasound	58.4 (6.3)	1331 (3.0)	7 (0.5)	1.4 (0.5)	1133 (2.9)	17 (1.5)	1331 (3.0)	56 (4.2)	11.6 (1.5)
	Group 7: Only additional mammography	58.8 (6.2)	217 (0.5)	0	0	185 (0.5)	<5 (<2.7)	217 (0.5)	5 (2.3)	11.7 (1.2)

CI: Cumulative incidence. IC, Interval cancer. BC, Breast cancer. NA, not applicable. *N***: Of 44,268 women, 37,964 participated in a second mammography screening, leaving 37,964 women for estimation of the risk of next-round screen-detected cancer. *N***: A total of 44,268 women with false-positive mammograms were included to estimate interval cancer and cumulative risks. Mean (SD): the mean and standard deviation of age in years. Mean (SD) Time to IC in years: the mean and standard deviation of time to interval cancer in years. Mean (SD) Follow-up time to BC in years: the mean and standard deviation of long-term breast cancer follow-up time in years. Groups 2, 3, and 4 are combined in the columns ‘Interval cancer’ and ‘Next-round Screen-detected cancer’ due to small numbers. <5: numbers less than 5 are not reported to comply with the General Data Protection Regulation and the Danish Health Data Authority [23].

**Table 2 cancers-15-01867-t002:** The crude and adjusted relative risks of interval cancer after false-positive mammograms by diagnostic assessment groups.

	Interval Cancer	RR, Crude	95% Confidence Interval	RR, Adjusted	95% Confidence Interval
Invasive diagnostic assessments	Group 1: All three elements (Standard of care)	Ref			
	Group 2: Only biopsy	0.63	(0.09–4.47)	0.62	(0.09–4.41)
	Group 3: Biopsy and additional mammography	1.12	(0.16–7.96)	1.09	(0.15–7.72)
	Group 4: Biopsy and ultrasound	0.34	(0.05–2.45)	0.32	(0.05–2.32)
Non-invasive diagnostic assessments	Group 5: Ultrasound and additional mammography (Standard of care)	Ref			
	Group 6: Only ultrasound	0.99	(0.47–2.11)	0.96	(0.45–2.04)
	Group 7: Only additional mammography	NA			
Both invasive and non-invasive assessments references	Group1: All three elements	Ref			
	Group 5: Ultrasound and additional mammography	0.67	(0.51–0.87)	0.66	(0.51–0.86)

RR, relative risk. NA, not applicable. Ref: the reference group. Adjusted RR: the relative risk of interval cancer adjusted for age groups.

**Table 3 cancers-15-01867-t003:** The crude and the adjusted relative risk of next-round screen-detected cancer after false-positive mammograms by diagnostic assessment groups.

	Next-Round Screen-Detected Cancer	RR, Crude	95% Confidence Interval	RR, Adjusted	95% Confidence Interval
Invasive diagnostic assessments	Group 1: All three elements (Standard of care)	Ref			
	Group 2: Only biopsy	0.98	(0.24–3.92)	0.98	(0.24–3.92)
	Group 3: Biopsy and additional mammography	NA			
	Group 4: Biopsy and ultrasound	1.32	(0.54–3.22)	1.22	(0.50–2.96)
Non-invasive diagnostic assessments	Group 5: Ultrasound and additional mammography (Standard of care)	Ref			
	Group 6: Only ultrasound	1.61	(0.99–2.62)	1.52	(0.93–2.47)
	Group 7: Only additional mammography	1.74	(0.56–5.37)	1.67	(0.54–5.16)
Both invasive and non-invasive assessments references	Group1: All three elements	Ref			
	Group 5: Ultrasound and additional mammography	0.80	(0.64–1.01)	0.78	(0.62–0.98)

RR, relative risk. NA, not applicable. Ref: the reference group. Adjusted RR: the relative risk of next-round screen-detected cancer adjusted for age groups.

**Table 4 cancers-15-01867-t004:** The hazard ratio of a long-term breast cancers risk after false-positive mammograms by diagnostic assessment groups.

	Long-Term Breast Cancers until End of 2021	HR, Crude	95% Confidence Interval	HR, Adjusted	95% Confidence Interval
Invasive diagnostic assessments	Group 1: All three elements (Standard of care)	Ref			
	Group 2: Only biopsy	0.72	(0.32–1.60)	0.72	(0.32–1.62)
	Group 3: Biopsy and additional mammography	0.73	(0.27–1.96)	0.76	(0.28–2.02)
	Group 4: Biopsy and ultrasound	0.75	(0.44–1.27)	0.75	(0.44–1.27)
Non-invasive diagnostic assessments	Group 5: Ultrasound and additional mammography (Standard of care)	Ref			
	Group 6: Only ultrasound	0.94	(0.72–1.22)	0.94	(0.72–1.23)
	Group 7: Only additional mammography	0.45	(0.19–1.09)	0.46	(0.19–1.10)
Both invasive and non-invasive assessments references	Group1: All three elements	Ref			
	Group 5: Ultrasound and additional mammography	0.92	(0.83–1.02)	0.92	(0.83–1.02)

HR, hazard ratio. Ref: the reference group. Adjusted HR: the hazard ratio of long-term breast cancer adjusted for age groups.

## Data Availability

The data that support the findings of this study are available through The Danish Health Data Authority. Restrictions apply to the availability of these data, which were used under license for this study. Data may be available upon reasonable request to The Danish Health Data Authority [29].

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
