# Peer review of "Choice of Assessment and Subsequent Risk of Breast Cancer among Women with False-Positive Mammography Screening"

_cancers, 2023, doi:10.3390/cancers15061867_

Round 1
Reviewer 1 Report
Overall comments
This study reports the incidence of interval cancers, subsequent screen-detected cancers, and any cancer diagnosis for patients after a false-positive mammogram receiving different levels of care. The results seem to justify the existing European guidelines for screening after a suspicious finding on mammogram. While novelty is limited, I believe that this study will be of interest to clinicians and researchers interested in the management of false-positive findings on mammogram. Below, I’ve provided some comments that might improve the manuscript.
Abstract
The abstract was difficult to understand. I would recommend clearly laying out the study population, the reference group (i.e. ‘standard of care’ with all three elements), and stating the authors’ interpretation of the study results.
Introduction
I found the purpose of this study to be difficult to understand. The final line seems to suggest that you are investigating a causal link between additional imaging itself and the risk of breast cancer (use of word “depends”). In reality, it seems to me that you are assessing whether women who don’t receive recommended assessment have an increased risk of missed cancers?
Minor comments: “solemnly” should be “solely”; Line 71, “Furthermore…” is not a complete sentence.
Methods
Please justify restriction to ages 49-71.
It would be helpful to elaborate on the exclusion of women with a prior history of breast cancer.
Minor comments: “remove “…, hence the quality is considered high.” Could put in Strengths and Weaknesses.
Line 126. I found it surprising that MRI was grouped with mammogram. Can you justify this? MRI is way more sensitive than mammogram and I wonder if results would differ. Also, were any of the mammograms contrast-enhanced?
Under “Analyses”: do you consider “(a)” to be an evaluation of misclassification and “(b)” to be an evaluation of inherent risk associated with false positive findings? That is my understanding.
Results
Statistical significance of measures is presented, but non-significant results are also presented as findings. See, e.g. lines paragraph starting on line 210. Is statistical significance taken into account or not?
Table 1 was difficult to understand. Suggest labeling “Incidence” as “Cumulative Incidence”. A footnote explaining the use of “<” in the results would be useful for readers not familiar with European data privacy rules. “Right cell-wise proportion” isn’t intuitive to me.
Please report the numbers of loss-to-follow-up and deaths.
Line 210. I would consider these results ‘null’.
Line 215. This is the only statistically significant finding, correct? Rephrase to something like, “For women who received both ultrasound and mammogram, those who did not receive a biopsy had an increased risk of …”. If I understand correctly.
Discussion
When rephrasing “main findings” it would be easier to understand if you included the clinical implications in the same section. It is hard to understand the key takeaways from the paper. Do your results support the current recommendations? Do the results suggest that some women who do not receive biopsies should receive biopsies?
Line 309: remove “small fraction” and replace with something more quantitative from your results.
Minor: “wherefore” I think should be changed to “therefore.”
Author Response
Comments to the editor: ’Thanks for the opportunity to revise the manuscript entitled “Choice of assessment and subsequent risk of breast cancer among women with false-positive mammography screening” Manuscript ID: cancers-2206758.
Changes to the manuscript are highlighted in the revised version.
We have added further references to comply with the journal instructions
Comments to the reviewers:
Thank you for giving us an opportunity to review this manuscript (cancers-2206758). In this study, we sought to investigate whether there was an association between the diagnostic assessments and subsequent breast cancer among women with false-positive mammography screening. Our responses to your concerns are detailed below. Changes to the manuscript are highlighted in yellow.
Reviewer1:
The abstract
#1
The abstract was difficult to understand. I would recommend clearly laying out the study population, the reference group (i.e. ‘standard of care’ with all three elements), and stating the authors’ interpretation of the study results.
Response: We thank the reviewer for the suggestions that have certainly improved the abstract. We have now clarified how the study population is defined as well as included a short description of 'standard of care', which includes two groups (Ref1: Additional mammography, ultrasound and biopsy) and in case of no need for an invasive procedure (Ref2: Additional mammography and ultrasound). We have also stated our interpretation of the results.
The revised sentences in the abstract now read:
This study aimed to investigate if the subsequent risk of breast cancer after a false-positive mammography screening is associated with the received diagnostic assessment.
The study population consist of women with false-positive mammography screening from January 2010 through June 2019. They were categorised into seven groups depending on the elements in the assessment (Standard care: additional mammography, ultrasound, and if relevant biopsy).
Risks of interval cancer, next-round screen-detected cancer, and long-term breast cancer for non-standard care assessments were compared to standard care assessments using Binomial and Cox regression models.
The increased subsequent risk of breast cancer among women with a previous false-positive result can only in a few cases be explained by choice of elements in the assessment process.
Introduction:
#2
I found the purpose of this study to be difficult to understand. The final line seems to suggest that you are investigating a causal link between additional imaging itself and the risk of breast cancer (use of word “depends”). In reality, it seems to me that you are assessing whether women who don’t receive recommended assessment have an increased risk of missed cancers?
Minor comments: “solemnly” should be “solely”; Line 71, “Furthermore…” is not a complete sentence.
Response: We acknowledge that the description of the purpose was unclear. The description has been changed to “This study aimed to investigate whether women who have had a false-positive mammography screening and did not receive recommended assessment have an increased subsequent risk of being diagnosed with breast cancer”. We have also changed “depends” in line 77 to “associated with”. We have as well corrected the minor mistakes with solely in both places in the article and completed the sentence on line 72-73 in the Introduction: “Furthermore, if relevant, biopsy of the detected lesion should be performed”.
Methods
#3
Please justify restriction to ages 49-71. It would be helpful to elaborate on the exclusion of women with a prior history of breast cancer.
Minor comments: “remove “…, hence the quality is considered high.” Could put in Strengths and Weaknesses.
Response: We have elaborated on the restriction to ages 49-71 as well as on the exclusion of women with a prior history of breast cancer. The sentences now read: “Although the Danish screening age targets women aged 50-69 years, some women are invited just before turning 50 and some have their last invitation at age 71, [13, 14] the study population therefore consisted of all women invited at age 49-71 years, who had a false-positive result following a mammography screening invitation between January 1st 2010 and June 30th 2019. Women who have previously been diagnosed with breast cancer were excluded from the study population as they have a higher risk of interval cancers than women without a previous breast cancer diagnose [15]. As we also expect that the majority of women with previous breast cancer would have received the standard of care assessment this would have biased our results.”
We have removed “hence the quality is considered high” from the Methods section and have included "DKMS obtains the data directly from the regions responsible for inviting the aimed population, hence the quality is considered high.” to the Strength and Weaknesses section, line 278-280.
#4
Line 126. I found it surprising that MRI was grouped with mammogram. Can you justify this? MRI is way more sensitive than mammogram and I wonder if results would differ. Also, were any of the mammograms contrast-enhanced?
Response: We thank the reviewer for raising this important point and we fully agree that it could be very interesting to stratify further both in terms of MRIs, tomosyntheses, or whether mammograms were contrast-enhanced. However, our main objective was to investigate whether women who did not receive the recommended assessment have an increased subsequent risk of being diagnosed with breast cancer compared to women who receive the recommended assessment (see reviewer 2, #1). As the recommended assessment includes 1) supplementary mammographic images, 2) ultrasound and if relevant 3) biopsy, we did not divide mammographic images into subcategories as e.g. tomosynthesis and MRI. Moreover, dividing into further groups to which MRI and tomosynthesis are already added would divide our relatively small study population into +20 groups, this will also lead to even wider confidence intervals and reduce the power of the study. In addition, due to GDPR and the Danish Health Data Authority, we are not allowed to report the numbers of all these groups since most of them will be below 5 (See #7), and this would also make the results non-applicable and hence less informative.
However, as we do agree that this is very interesting to investigate, we added the following to the discussion: “Finally, according to guidelines, examinations such as tomosynthesis and MRI fall under the category 'supplementary mammographic examinations'. As these procedures have a higher sensitivity detecting breast cancer, it could have been interesting to look at them separately. However, this was not possible due to small numbers.”
As for contrast-enhancement for mammography, this has not been and is still not recommended in Danish national guidelines for diagnostic assessment within screening, and none of the mammograms were to our knowledge contrast-enhanced.
#5
Under “Analyses”: do you consider “(a)” to be an evaluation of misclassification and “(b)” to be an evaluation of inherent risk associated with false positive findings? That is my understanding.
Response: We thank the reviewer for pointing out that this should be explained better.. It’s now clarified under Analyses as “To investigate the risk of misclassification, we calculated for each assessment group a) the risk of getting an interval cancer in the two years following the false-positive screening and b) the risk of getting screen-detected cancer diagnosed at the successive screening (next-round). Furthermore, to investigate how different choices of assessment correlate with the subsequent risk of breast cancer, we calculated for each assessment group c) the long-term risk of getting breast cancer any time before end of follow-up, i.e. including both a, b, and breast cancers diagnosed due to symptoms or at later screening examinations.”
Results:
#6
Statistical significance of measures is presented, but non-significant results are also presented as findings. See, e.g. lines paragraph starting on line 210. Is statistical significance taken into account or not?
Response: We do agree that the results are not statistically significant, but that does not exclude that the estimates are clinically relevant. We acknowledge, that this should be elaborated on in the discussion, which is included in the revised manuscript line 270-274:
“Although the statistical significance is absent in the results due to the few numbers in the groups with ‘only ultrasound’ and ‘only additional mammography’ assessments, the clinical significance might be present. We therefore reported them as point and confidence interval estimates as recommended by STROBE and others [24-26].”
#7
Table 1 was difficult to understand. Suggest labeling “Incidence” as “Cumulative Incidence”. A footnote explaining the use of “<” in the results would be useful for readers not familiar with European data privacy rules. “Right cell-wise proportion” isn’t intuitive to me.
Please report the numbers of loss-to-follow-up and deaths.
Response: We thank the reviewer for pointing this out. We fully agree and changes were made accordingly to table 1. A footnote is added to table 1 explaining that “<5: Numbers is less than 5 and is not applicable according to General Data Protection Regulation and the Danish Health Data Authority [23].” And “Right cell-wise proportion” is now removed from table 1 footnotes to annul the confusion.
The numbers of loss-to-follow-up and deaths is now added in the results as “The survival analysis for long-term breast cancer shows that 2,020 (4.6%) were censored due to death and none due to emigration.”
#8
Line 210. I would consider these results ‘null’.
Response: See response #6
#9
Line 215. This is the only statistically significant finding, correct? Rephrase to something like, “For women who received both ultrasound and mammogram, those who did not receive a biopsy had an increased risk of …”. If I understand correctly.
Response: We are in this paragraph comparing the 2 standard of care assessments, i.e. 'ultrasound & additional mammography' to the ‘all three elements' group. The results showed significantly lower risks of getting an interval cancer and next-round screen-detected cancer, and a non-significant lower risk of getting long-term breast cancer with a follow-up of 12 years (Table 2-4). This is not surprising as women referred to a biopsy would most likely have a lump, whereas women not referred to biopsy would most likely not have a lump. We acknowledge that we need to emphasize this in the discussion section and have added this in line 328-330: “In addition, our results as well as the Spanish results that women referred to assessment including a biopsy have a higher risk of breast cancer are as expected since women referred to a biopsy would most probably have a lesion.”
Discussion
#10
When rephrasing “main findings” it would be easier to understand if you included the clinical implications in the same section. It is hard to understand the key takeaways from the paper.
Response: We agree with the reviewer on including the clinical implications in the main findings section. It now reads “Main findings
This study showed that women who received invasive diagnostic follow-up without all three elements (i.e. additional mammography, ultrasound, and biopsy) had a lower or similar risk of interval cancer, next-round screen-detected cancer, and long-term breast cancer compared to women who received all three elements, therefore, missing one or two elements other than the biopsy element do not seem to be problematic. Women who had only ultrasound or only additional mammography during a non-invasive assessment had a non-statistically significant higher risk of next-round screen-detected cancers compared to women who received both assessments combined, hence missing one of the elements might be related to a missed cancer.”
#11
Do your results support the current recommendations? Do the results suggest that some women who do not receive biopsies should receive biopsies?
Response: We agree that this is a very interesting question. However, this study is not designed to evaluate the recommendations. The results though support the current practice, where very few deviate from the standard of care and they do not seem to constitute a higher risk. The invasive and non-invasive groups differ from each other in many aspects, e.g. biological features. Therefore, the study has two reference groups to be compared with their almost-similar groups.
#12
Line 309: remove “small fraction” and replace with something more quantitative from your results.
Response: Thanks for pointing that out. However, we cannot provide a specific number, because there are too few cases (See response #7) and the risk is not that much elevated. It now reads “The risk of interval cancer, next-round screen-detected cancer, and long-term breast cancer in groups with varying combinations of assessments showed that the increased subsequent risk of breast cancer among women with a previous false-positive result can only in a few cases be explained by choice of elements in the assessment process.”
#13
Minor: “wherefore” I think should be changed to “therefore.”
Response: We fully agree and we changed the word in the text.
Reviewer2:
#1
Is there any literature to suggest that certain types of assessment may lead to an increased risk of breast cancer, or do you have any hypotheses as to why particular modalities may or may not increase the risk of breast cancer? i.e. is there a biological basis?
Response: Thanks for the good question. The project was conceptualized, as we found it interesting to investigate whether assessments that did not follow guidelines recommendations were followed by an increased risk of breast cancer. We hope that this is clearer now that the last paragraph of the introduction reads: This study aimed to investigate whether women who have had a false-positive mammography screening and did not receive recommended assessment have an increased subsequent risk of being diagnosed with breast cancer.
#2
Additional mammography is defined as including tomosynthesis and breast MRI – aren’t these methods likely to differ in the rates of diagnosis of cancer and therefore also impact on the diagnosis of interval and next-screen detected breast cancers? Why report ultrasound separately and not DBT and MRI? I realise some of the numbers are likely to be small but the number receiving each of these modalities could be reported before combining into one category.
Response: See response #4 to reviewer 1
#3
Did you consider looking at the differences according to whether screens were incident or prevalent – again I acknowledge the numbers in some groups may be small, but again this could be reported. Where previous mammograms used in forming a judgement as to the type of assessment? – both of these points should be addressed in the discussion.
#4
Can you provide more information on the demographics of the study population e.g. age (as a minimum) for each category? Can you also provide the mean (SD)/median(IQR) follow-up time for the long term breast cancers, and mean (SD)/median(IQR) time to interval cancer.
Response: We have added mean (SD) of age, mean (SD) of follow-up time for the long term breast cancers, and mean (SD) of time to interval cancer to table1.
#5
Confidence intervals on the whole were wide – some groups were based on relatively small numbers of cancers, please add this to the discussion section.
Response:
Response: We fully agree and we have elaborated on that in the discussion as “Although the statistical significance is absent in the results due to the few numbers of whom were assessed in the ‘only ultrasound’ and the ‘only additional mammography’ assessments, the clinical significance is present and recommended to be reported as point and interval estimates by STROBE and others [24-26].”. See also answer to Reviewer 1, question 6 & 8.
#6
What is included in the adjusted model – is it the model with age and the age and group interaction term? – please clarify.
Response: Thanks for pointing out that this information was lacking. It is now added to the method section as “A test for interaction between age and the assessment groups was performed in all three models. The interaction terms were then dropped and the adjusted models included the age groups only together with the assessment groups”
#7
There may be different reasons for assessment with and without biopsy e.g. non-invasive assessment may be performed in those with high breast density, whereas biopsy may be performed due to other suspicious features on the mammogram e.g. presence of micro-calcifications, asymmetry etc.? - you should address this within the discussion section. In addition, there may be other confounding factors. Similarly, choice of non-invasive assessment (e.g. DBT, MRI and ultrasound) may have been influenced by mammographic features.
#8
It may be my computer but the flow chart seems to have a black background? Add percentages for other exclusions?
Response: Thanks for pointing that out. In our versions it’s not black, but we changed it to a PNG format, hoping that it is visually improved. We will as well pay attention to that when going through the proofs.
#9
Reporting percentages to one decimal place is probably sufficient.
Response: Thanks for your comment. We have changed to one decimal in table 1 but not in the other tables as we would. rather not substitute .99 to 1.
#10
-Table 1: are groups 2, 3 & 4 combined – or is it just group 3 you have reported and Groups 2 and 4 were 0? I assume it’s the former but I’d suggest adding a footnote for clarification.
-Why are some groups <5? - is this for the purposes of suppressing small numbers in those with <5 cancers – if so, please add a footnote to say this.
Response: We agree that a footnote is needed and it’s now added as a footnote under table1 as
“-The groups 2, 3 and 4 are combined in the columns ‘Interval cancer’ and ‘Next-round Screen-detected cancer’, due to the very few numbers.
-<5: Numbers is less than 5 and is not applicable according to General Data Protection Regulation and the Danish Health Data Authority [23]”
#11
Group 0: no follow-up – should this be ‘no assessment’ rather than ‘No follow-up’ as they still have long term outcomes?
Some of the text, particularly in the discussion section, could be more clearly written e.g.
Page 3, line 95 – remove one of the instances of ‘if relevant’
Page 4, line 170 – delete the first two words ‘samples from’
Page 6, line 190 – change decimal point to comma in 26.655 and comma to decimal point in 68,91%
You use the term ‘solemnly explain’ in the intro (page 2, line 63) & discussion (page 10, 292) but perhaps you mean ‘solely explain’?
Response: Thanks for pointing the above points out. We fully agree with the reviewer and all the above changes is now added to/ removed from table1 and the text.

Reviewer 2 Report
This paper reports on subsequent risk of breast cancer in women with a false-positive mammography screening and the relationship with type of assessment (e.g. ultrasound, biopsy).
Is there any literature to suggest that certain types of assessment may lead to an increased risk of breast cancer, or do you have any hypotheses as to why particular modalities may or may not increase the risk of breast cancer? i.e. is there a biological basis?
Additional mammography is defined as including tomosynthesis and breast MRI – aren’t these methods likely to differ in the rates of diagnosis of cancer and therefore also impact on the diagnosis of interval and next-screen detected breast cancers? Why report ultrasound separately and not DBT and MRI? I realise some of the numbers are likely to be small but the number receiving each of these modalities could be reported before combining into one category.
Did you consider looking at the differences according to whether screens were incident or prevalent – again I acknowledge the numbers in some groups may be small, but again this could be reported. Where previous mammograms used in forming a judgement as to the type of assessment? – both of these points should be addressed in the discussion.
Can you provide more information on the demographics of the study population e.g. age (as a minimum) for each category? Can you also provide the mean (SD)/median(IQR) follow-up time for the long term breast cancers, and mean (SD)/median(IQR) time to interval cancer.
Confidence intervals on the whole were wide – some groups were based on relatively small numbers of cancers, please add this to the discussion section.
What is included in the adjusted model – is it the model with age and the age and group interaction term? – please clarify.
There may be different reasons for assessment with and without biopsy e.g. non-invasive assessment may be performed in those with high breast density, whereas biopsy may be performed due to other suspicious features on the mammogram e.g. presence of microcalcifications, asymmetry etc? - you should address this within the discussion section. In addition, there may be other confounding factors. Similarly, choice of non-invasive assessment (e.g. DBT, MRI and ultrasound) may have been influenced by mammographic features.
Minor comments
It may be my computer but the flow chart seems to have a black background? Add percentages for other exclusions?
Reporting percentages to one decimal place is probably sufficient.
Group 0: no follow-up – should this be ‘no assessment’ rather than ‘No follow-up’ as they still have long term outcomes?
Table 1: are groups 2, 3 & 4 combined – or is it just group 3 you have reported and Groups 2 and 4 were 0? I assume it’s the former but I’d suggest adding a footnote for clarification.
Why are some groups <5? - is this for the purposes of supressing small numbers in those with <5 cancers – if so, please add a footnote to say this.
Some of the text, particularly in the discussion section, could be more clearly written e.g.
Page 3, line 95 – remove one of the instances of ‘if relevant’
Page 4, line 170 – delete the first two words ‘samples from’
Page 6, line 190 – change decimal point to comma in 26.655 and comma to decimal point in 68,91%
You use the term ‘solemnly explain’ in the intro (page 2, line 63) & discussion (page 10, 292) but perhaps you mean ‘solely explain’?
Author Response

(The authors gave the same response as above.)

Round 2
Reviewer 1 Report
The authors have carefully responded to my suggestions. I still found the purpose of the paper in the Abstract and Introduction difficult to follow. The revised last paragraph of the Introduction better captures the purpose, but the comparison groups need to be better described in the Methods. A few small suggestions.
Abstract. The revised conclusion "The increased subsequent risk of...can only in a few cases be explained by ..." is not a clear conclusion.
Methods. I did not find the exclusion of women with a prior breast cancer convincing. I don't see how "bias" would result due to the women with a previous breast cancer receiving standard of care. Indeed, I think that a lot of the population with false-positive results have a prior history of cancer, so this could improve generalizability.
Discussion.
Response to #10. The use of the word "problematic" as a conclusion is again vague and needs to be clarified.
Author Response
We thank the reviewers again for the comments, and appreciate your aim of improving the quality of the article. Our response is giving point by point manner below, and revised in the manuscript accordingly.
Reviewer #1
#1
The authors have carefully responded to my suggestions. I still found the purpose of the paper in the Abstract and Introduction difficult to follow. The revised last paragraph of the Introduction better captures the purpose
Response: We have revised the aim of the paper according to the recommendation of reviewer #2 and the revised purpose in the introduction now reads. “This study aimed to investigate whether women who had a false-positive mammography screening and did not receive recommended assessment as per guidelines had an increased risk of being diagnosed with a subsequent breast cancer”.
#2
The comparison groups need to be better described in the Methods.
Response: We thank the reviewer for the suggestion. The groups in the method section now categorized as : Standard of care, i.e. additional mammography, ultrasound, and biopsy (group 1), assessment including only biopsy (group 2), assessment including biopsy and additional mammography (group 3), and assessment including biopsy and ultrasound (group 4). Similarly, the non-invasive diagnostic assessments were grouped into standard of care, i.e. additional mammography and ultrasound (group 5), assessment including only ultrasound (group 6), and assessment including only additional mammography (group 7).
A few small suggestions.
Abstract. The revised conclusion "The increased subsequent risk of...can only in a few cases be explained by ..." is not a clear conclusion.
Response: We have revised the sentence and it reads as following both in the abstract and the conclusion, respectively.
- “The increased subsequent risk of breast cancer among women with a previous false-positive result was not found to be correlated with the choice of elements in the assessment process.”
- “The risk of interval cancer, next-round screen-detected cancer, and long-term breast cancer in women with false-positive mammography screening was not higher in groups with varying combinations of assessments compared to standard of care. Thus, the increased subsequent risk of breast cancer among women with a previous false-positive result may not be explained by choice of elements in the assessment process”
Methods. I did not find the exclusion of women with a prior breast cancer convincing. I don't see how "bias" would result due to the women with a previous breast cancer receiving standard of care. Indeed, I think that a lot of the population with false-positive results have a prior history of cancer, so this could improve generalizability.
Response: We thank the reviewer for the suggestions, but we still think that including the previously diagnosed women will skew our results as it is clarified in the following. We have though added the sentence highlighted to improve the clarity of our argument. The revised paragraph now reads “Women who had previously been diagnosed with breast cancer were excluded from the study population as they have a higher risk of interval cancers than women without a previous breast cancer diagnosis [15]. As we also expect that the majority of women with previous breast cancer would have received the most comprehensive follow-up, i.e. standard of care assessment, this would have biased our results, since including them could increase the risk of subsequent cancer in the standard of care assessment group.”
Discussion.
Response to #10. The use of the word "problematic" as a conclusion is again vague and needs to be clarified.
Response: We acknowledge that the word “problematic” does not provide a clear meaning. The sentence has been changed to “Therefore, missing one or two elements other than the biopsy does not seem to result in an increase of missed cancers”.

Reviewer 2 Report
Thank you, the authors have addressed my earlier comments, however I have made some minor suggestions and also some suggestions to help improve the clarity of certain aspects of the paper (see below).
Line 33: change ‘consists’ to ‘consisted’
Line 82-84: could be phrased more clearly e.g: ‘This study aimed to investigate whether women who have had a false-positive mammography screening and did not receive the recommended assessment as per guidelines hadve an increased subsequent risk of being diagnosed with a subsequent breast cancer.’
Line 104: change ‘invited at’ to ‘between’
Line 106: change ‘Women who have’ to ‘Women who had’
Line 173 -175: please clarify if models were adjusted using age as a continuous variable or in age groups?
Table 1 – please also add that the mean (SD) columns are the time to interval cancer and follow-up time (in years) for long-term breast cancers in the table as well as the footnote to avoid any confusion
Change the footnote as follows:
Groups 2, 3 and 4 are combined in columns ‘Interval cancer’ and ‘Next-round Screen-detected cancer’, due to small numbers.
<5: Numbers less than 5 are not reported to comply with….
Table 2,3 and 4 – please add a footnote to describe the difference between the unadjusted and adjusted models.
Line 262 – 266: could be phrased more clearly e.g. ‘This study showed that women who received invasive diagnostic follow-up without all three elements (i.e. additional mammography, ultrasound, and biopsy) had a lower or similar risk of interval cancer, next-round screen-detected cancer, and long-term breast cancer compared to women who received all three elements. , tTherefore, missing one or two assessments, elements other than the biopsy, element does not seem to be problematic.’
Line 270 -274 could be phrased more clearly e.g. ‘Although the statistical significance is absent in the results due to the fewsmall numbers in some the groups with only ultrasound and only additional mammography assessments, the clinical significance might may be present. We therefore reported them results as point estimates with and 95% confidence interval estimates as recommended by STROBE and others [24-26]’.
Line 290: change ‘wherefore’ to ‘therefore’
Line 300: change ‘remaining’ to ‘remainder’
Line 302: delete the word ‘though’
Line 303: change ‘wherefore’ to ‘therefore’
Lines 306 -312: this could be phrased more clearly e.g. ‘Choice of assessment as well as subsequent risk of breast cancer might be different for incident and prevalent screened women. However, as our data does not include the prevalent round of the Danish mammography screening programme, most prevalent screenings are in women aged 50-51 years, with limited number of subsequent breast cancers. For instance, our study population consisted ofcontained only 655 prevalent screened women who also had a next screening at whichwith 5 breast cancers were detected at their next screening round. These low numbers among prevalent screened women can will not have affected our results notably
Lines 316: change ‘higher sensitivity detecting breast cancer, it could have….’ to ‘higher sensitivity in detecting breast cancer, it would have….’
Line 328-330: this could be phrased more clearly e.g. ‘In addition, our results confer with the Spanish study in that as well as the Spanish results that women referred to assessment including a biopsy hadve a higher risk of breast cancer, as these is as expected since women referred to a biopsy, were more likely to would most probably have had a lesion.’
Line 333: change ‘referred to a biopsy’ to ‘referred for a biopsy’
Line 339: change ‘go in line’ to ‘is in line’
Line 348 -351: could be phrased more clearly e.g. ‘Moreover, other underlying reasons for the choice of examinations in the assessments may have been influenced by other factors , such as suspicious clinical findings or particular mammographic features which , should be considered when assessing the extent of confounding, althoughin further studies as this was beyond the scope of thisour study.’
Author Response
We thank the reviewers again for the comments, and appreciate your aim of improving the quality of the article. Our response is giving point by point manner below, and revised in the manuscript accordingly.
Reviewer #2
Line 33: change ‘consists’ to ‘consisted’
Line 104: change ‘invited at’ to ‘between’
Line 106: change ‘Women who have’ to ‘Women who had’
Line 173 -175: please clarify if models were adjusted using age as a continuous variable or in age groups?
Table 1 – please also add that the mean (SD) columns are the time to interval cancer and follow-up time (in years) for long-term breast cancers in the table as well as the footnote to avoid any confusion
Change the footnote as follows:
Groups 2, 3 and 4 are combined in columns ‘Interval cancer’ and ‘Next-round Screen-detected cancer’, due to small numbers.
<5: Numbers less than 5 are not reported to comply with….
Table 2,3 and 4 – please add a footnote to describe the difference between the unadjusted and adjusted models.
Line 290: change ‘wherefore’ to ‘therefore’
Line 300: change ‘remaining’ to ‘remainder’
Line 302: delete the word ‘though’
Line 303: change ‘wherefore’ to ‘therefore’
Lines 316: change ‘higher sensitivity detecting breast cancer, it could have….’ to ‘higher sensitivity in detecting breast cancer, it would have….’
Line 333: change ‘referred to a biopsy’ to ‘referred for a biopsy’
Line 339: change ‘go in line’ to ‘is in line’
Line 82-84: could be phrased more clearly e.g: ‘This study aimed to investigate whether women who have had a false-positive mammography screening and did not receive the recommended assessment as per guidelines hadve an increased subsequent risk of being diagnosed with a subsequent breast cancer.’
Line 262 – 266: could be phrased more clearly e.g. ‘This study showed that women who received invasive diagnostic follow-up without all three elements (i.e. additional mammography, ultrasound, and biopsy) had a lower or similar risk of interval cancer, next-round screen-detected cancer, and long-term breast cancer compared to women who received all three elements. , tTherefore, missing one or two assessments, elements other than the biopsy, element does not seem to lead to missed cancers.’
Line 270 -274 could be phrased more clearly e.g. ‘Although the statistical significance is absent in the results due to the fewsmall numbers in some the groups with only ultrasound and only additional mammography assessments, the clinical significance might may be present. We therefore reported them results as point estimates with and 95% confidence interval estimates as recommended by STROBE and others [24-26]’.
Lines 306 -312: this could be phrased more clearly e.g. ‘Choice of assessment as well as subsequent risk of breast cancer might be different for incident and prevalent screened women. However, as our data does not include the prevalent round of the Danish mammography screening programme, most prevalent screenings are in women aged 50-51 years, with limited number of subsequent breast cancers. For instance, our study population consisted ofcontained only 655 prevalent screened women who also had a next screening at whichwith 5 breast cancers were detected at their next screening round. These low numbers among prevalent screened women can will not have affected our results notably
Line 328-330: this could be phrased more clearly e.g. ‘In addition, our results confer with the Spanish study in that as well as the Spanish results that women referred to assessment including a biopsy hadve a higher risk of breast cancer, as these is as expected since women referred to a biopsy, were more likely to would most probably have had a lesion.’
Line 348 -351: could be phrased more clearly e.g. ‘Moreover, other underlying reasons for the choice of examinations in the assessments may have been influenced by other factors, such as suspicious clinical findings or particular mammographic features which, should be considered when assessing the extent of confounding, althoughin further studies as this was beyond the scope of thisour study.’
Response: We have implemented all the reviewer’s suggestions into the article. And we thank the reviewer for the suggestions that have certainly improved the clarity of the article.
